

# Prevalence and associated risk factors of avian influenza A virus subtypes H5N1 and H9N2 in LBMs of East Java province, Indonesia: a cross-sectional study

Saifur Rehman[1,2,3], Mustofa Helmi Effendi[1], Aamir Shehzad[2], Attaur Rahman[4], Jola Rahmahani[2], Adiana Mutamsari Witaningrum[1] and Muhammad Bilal[3]

[1] Division of Veterinary Public Health, Faculty of Veterinary Medicine, Airlangga University, Surabaya, East Java, Indonesia
[2] Laboratory of Virology and Immunology Division of Microbiology, Faculty of Veterinary Medicine, Airlangga University, Surabaya, East Java, Indonesia
[3] Department of Epidemiology and Public Health, Faculty of Veterinary Science, University of Veterinary and Animal Sciences, Lahore, Punjab, Pakistan
[4] Department of Medicine and Therapeutics, The Chinese University of Hong Kong, SAR

Corresponding author
Mustofa Helmi Effendi,
mhelmieffendi@gmail.com

## ABSTRACT

**Background.** Avian influenza A virus subtypes H5N1 and H9N2 are contagious zoonotic diseases that are circulating in Indonesia and have raised increasing concern about their potential impacts on poultry and public health. A cross-sectional study was carried out to investigate the prevalence and associated risk factors of avian influenza A virus subtypes H5N1 and H9N2 among poultry in the live bird markets of four cities in East Java province, Indonesia.

**Methods.** A total of 600 tracheal and cloacal swabs (267 from backyards, 179 from broilers, and 154 from layers) from healthy birds were collected. The samples were inoculated into specific pathogenic-free embryonated eggs at 9-day-old *via* the allantoic cavity. qRT-PCR was used for further identification of avian influenza.

**Results.** The overall prevalence of circulating influenza A virus subtypes H5N1 and H9N2 was 3.8% (23/600, 95%CI [0.0229–0.0537]). Prevalence was higher in backyards at 5.99% (16/267) followed by broilers (2.23% (4/179)) and layers (1.68% (3/154)). The final multivariable model revealed five risk factors for H9N2 infections: presence of ducks ($p = 0.003$, OR = 38.2), turkeys ($p = 0.017$ OR = 0.032), and pheasants in the stall ($p = 0.04$, OR = 18.422), dry ($p = 0.006$) and rainy season ($p < 0.001$), and household birds ($p = 0.002$) and seven factors for H5N1 infections including: observing rodents ($p = 0.036$, OR = 0.005), stray dogs access ($p = 0.004$ OR $\leq$ 0.001), presence of turkeys ($p = 0.03$ OR = 0.007), chukars/partridges ($p = 0.024$ OR = 2500), and peafowls in the stalls ($p = 0.0043$ OR $\leq$ 0.001), rainy season ($p = 0.001$) and birds from the household sources ($p = 0.002$) in the live bird markets.

**Conclusions.** The findings of the current study illustrate the recurring infection and presence of both avian influenza viruses and associated risk factors in the surveyed marketplaces. Effective protective measures and mitigation strategies for risks outlined in this study could help to reduce the burden of H5N1 and H9N2 AI subtypes into the live bird markets of Indonesia.

## INTRODUCTION

Avian influenza (AI) is a highly contagious infection caused by one of the subtypes of influenza viruses that primarily infect birds and mammals, and are categorized into three subtypes (A, B, and C) based on nucleoprotein (NP) and matrix protein (MP) content (*Fouchier et al., 2005*). In Indonesia, the high pathogenic avian influenza A subtype H5N1 virus has been endemic since 2003, while the low pathogenic avian influenza A H9N2 virus was first reported in 2017 in the poultry population (*Jonas et al., 2018*; *Smith et al., 2006*). In addition to having a negative impact on poultry production, these viruses have caused sporadic influenza infections in humans as well (*Chakraborty, Arifeen & Streafield, 2011*; *World Health Organization, 2017*). Their continued recurrence in poultry poses a continuous severe threat to animal and human health globally due to the possibility of novel reassortment variations between them or with other virus subtypes . (*Chen et al., 2014*; *Gao et al., 2013*; *Lee et al., 2017*; *Lin et al., 2000*). In several Asian countries, live bird markets (LBMs) are the backbone of the poultry trade. Different types of birds from various geographic areas are introduced into LBMs daily and may be confined together, encouraging the local spread of several virus subtypes and allowing for reassortment (*Fournié et al., 2011*; *Nguyen et al., 2005*; *Webster, 2004*). Avian influenza surveys and regular monitoring among endemic countries, including Indonesia, have revealed the incidence and diversity of avian influenza A viruses (AIVs) in LBMs (*Chen et al., 2016*; *Dharmayanti et al., 2020*; *Huang et al., 2015*; *Liu et al., 2003*; *Okamatsu et al., 2013*; *Phan et al., 2013*). The hierarchical data structure particularly the grouping of sampled birds per LBMs are often ignored, and only the fraction of positive samples is reported. As a result, there is no credible estimate of AIV prevalence in LBMs, though the information is essential for understanding AIVs epidemiology and enhancing surveillance strategies (*Kim et al., 2018*).

Studies conducted in Hong Kong, China, Indonesia, and the United States revealed that live bird markets (LBMs) can carry avian influenza A virus subtypes H5N1 and H9N2 particularly highly pathogenic avian influenza A virus subtype H5N1, and can serve as potential sources of avian influenza A virus subtypes H5N1 and H9N2 transmission to humans (*Guan et al., 2007*). AIVs can be introduced, entrenched, and spread through the frequent movement of birds into, through, and out of marketplaces. Most investigations in the LBMs have focused on live birds rather than ambient settings (*Cardona, Yee & Carpenter, 2009*; *Garber et al., 2007*; *Wang et al., 2006*). LBMs in Indonesia continue to be a threat, as evidenced by the death of an 8-year-old child in West Java probably due to avian influenza on July 5, 2012, who visited LBMs with her father and brought freshly butchered birds home, as confirmed by the government officials (*Naysmith, 2014*).

Initial surveillance investigations in Indonesia found a significantly greater incidence of AIVs, particularly highly pathogenic avian influenza A virus subtype H5N1, at these LBMs than in poultry-producing areas, implying that the highly pathogenic avian influenza A virus subtype H5N1spreads widely during the trading process. Furthermore, there are significant

differences between the business model of domestic poultry (such as Kampung hen) and commercial poultry (such as broilers and layers) marketed in Indonesian urban LBMs (*Henning et al., 2019*). A study conducted in Indonesia on LBMs found that slaughtering of chickens at the LBMs increased the risk of avian influenza virus infections (*Indriani et al., 2010*).

LBMs may act as a reservoir for AIV, a serious threat to animal and human health, therefore, it is critical to identify local risk factors associated with the infection. The following are the most significant risk factors: placement of new birds with leftover birds from previous batches in the same cages (*Bulaga et al., 2003*; *Indriani et al., 2010*); mixing of different species in the same cages, sharing the slaughtering, (*Bulaga et al., 2003*); feeding, watering, and weighing equipment, and the presence of rats, etc. (*Kung, 2006*).

One of the most efficient strategies for preventing the spread of avian influenza (AI) viruses in Indonesia is the use of vaccinations (*Rehman et al., 2022a*; *Rehman et al., 2022b*). Vaccination against HPAI, which was heavily administered in Sector 3 layers in Indonesia, had extremely variable results. The Differentiating Infected from Vaccinated Animals (DIVA) technique, which is proposed to involve sentinel chickens, has been tested in West Java. In Indonesia, the DIVA strategy is not widely accepted as a solution to the problem (*Tarigan & Sciences, 2015*; *Bouma et al., 2008*). There have been a number of different strategies of viral proteins HA2, NS1, and M2e that have been developed in Indonesia that reduces the prevalence of AIVs in all poultry sectors (*Tumpey et al., 2005*; *Lee et al. , 2014*; *Suarez, 2012*).

The aim of this study was to determine the prevalence of avian influenza A virus subtypes H5N1 and H9N2 in LBMs in East Java, Indonesia, as well as to identify risk factors associated with the infection. This study was intended to help the concerned government authorities potentially involved in the control and eradication of avian influenza A virus subtypes H5N1 and H9N2 and its associated risk factors in the LBMs of Indonesia, specifically in the study province, as well as incorporate the findings of the study into future control and eradication strategies of this deadly infection.

## MATERIAL AND METHODS

### Ethical approval

The Animal Care and Use Committee at the Faculty of Veterinary Medicine, Universitas Airlangga Surabaya, Indonesia (approval no. 1.KE.028.03.2021) gave their approval for every step of this study.

### Study population and sampling

A cross-sectional study was designed from March 2021 to April 2022 to estimate the prevalence of avian influenza A virus subtypes H5N1 and H9N2 and the potential risk factors associated with the positivity among the poultry (backyard, layer, and broiler) of LBMs of four cities, including Surabaya, Sidoarjo, Pasuruan, and Malang in East Java, Indonesia (Fig. 1). A LBM was defined as an open space in which > 2 poultry stalls sell live poultry at least once a week, and only those selling >400 poultry per day were considered eligible for this study. The sample size was calculated using the Cochran formula with the

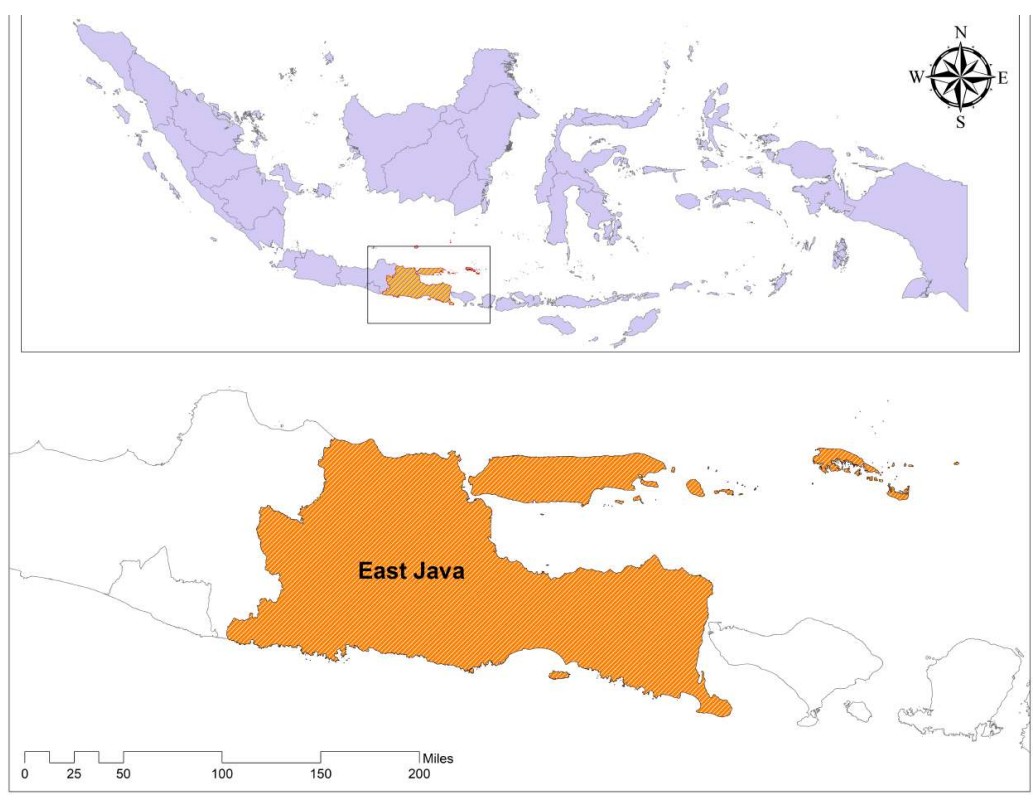

**Figure 1** Location of East Java in Indonesia.

population proportion assumed to be 50% with 95% confidence interval and 5% desired precision levels. The poultry stalls with more than 10 birds were selected for sampling. The birds that were used in the sampling came from a variety of sources, including farms (broiler and layer) and individual households (backyard). A total of 600 cloacal and tracheal swab samples were collected from healthy live birds having an age of more than two months from the LBMs of four cities (Surabaya = 150, Sidoarjo = 150, Malang = 150, and Pasuruan = 150). The sterilized swab was used to collect swab samples from the trachea and cloaca of the bird. The samples were placed in 3-ml transport media (phosphate-buffered saline (PBS) containing penicillin and streptomycin) in cryovials. The ice packs were used in an icebox to maintain the cold chain temperature of falcon tubes during transport. Afterward, the swab samples were stored at $-80C^0$ in the bio-molecular laboratory, Faculty of Veterinary Medicine, Universitas Airlangga Surabaya, Indonesia for further processing.

## Virus isolation

For virus isolation in eggs, 0.2–0.3 ml of undiluted oral or cloacal swab medium was inoculated into the allantoic cavity of 9–11-day-old embryonated specific-pathogen-free (SPF) chicken eggs (three eggs inoculated per sample). The eggs were sanitized with ethyl alcohol at a concentration of 70% before being inoculated. Antibiotics (penicillin and streptomycin) were also injected using a one mL sterile syringe to prevent the growth of

bacteria and contamination (*Krauss, Walker & Webster, 2012*). This process was followed by incubation at 37° C for 120 h (*Swayne, 1998*). Embryos were monitored every day for 3–4 days following inoculation to see if they died. Using normal techniques, allantoic secretions from all deceased eggs were analyzed for the presence of influenza virus using the HA test (*OIE, 2011*). H5N1 (clade. 2.3.2) and H9N2 specific standardized serum antibodies were used to assess HA positive samples by doing HI. The results of HI were compared to positive controls of avian influenza A virus subtypes H5N1 (clade 2.3.2.1c) and H9N2. The HI assay was utilized to determine the hemagglutinin (H) subtype of an unidentified AI virus isolate or the HA subtype specificity of AI virus antibodies. HI, positive samples were further processed to detect avian influenza virus with qRT- PCR.

## Detection of H5N1 and H9N2 using qRT-PCR

qRT-PCR was used to identify viral RNA in allantoic fluid. In brief, total RNA in the allantoic fluid was extracted according to the manufacturer's instructions using the QIAamp viral RNA Mini kit (QIAGEN, Hilden, Germany). The viral RNA copies were determined using the THUNDERBIRD™ probe one-step qRT-PCR kit (TOYOBO, New York, USA) with the designed primer and probes by Indonesia-Japan Collaborative Research Center for Emerging and Re-emerging Infectious Diseases, Institute of Tropical Disease University of Airlangga. The primer set for H5 (F, 5′-CGATCTAAATGGAGTGAAGCCTC-3′; R, 5′-CCTTCTCTACTATGT AAGACCATTC-3′) and Taqman probes (Thermo Fisher Scientific, Waltham, MA, USA) (FAM) FAMAGCCA TCCYG CTACA CTACA-MGB identifies the HA gene of European and Indonesian lineages of AI A virus subtype H5N1. We detected the M gene using following primers (F, 5′-CCMAG GTCGA AACGT AYGTT CTCTC TATC-3′; R, TGCAG RATYG GTCTT GTCTT TAGCC AYTCCA-3′) and probe (FAM-ATYTC GGCTT TGAGGGGGCCTG-MGB). The set of H9 primer (F, 5′-GGAAGAATTATTATTGGTCGGTAC-3′, R,5-′GCCACCTTTTTCAGTCTGACATT-3′), and the Taqman probe AACCAGGCCAGACATTGCGAGTAAGATCC[TAMRA] were taken from a previously published article by *El-Sayed et al. (2021)*. The reaction mixture consisted of 10 μl of reaction buffer, 0.5 μl DNA polymerase, 0.5 μl of RT enzyme, 1 μl of forwarding primer, 1 μl reverse primer, 0.8 μl probe, 1.2 μl RNAs free water, ROX dye, 0.04 μl, and 5 μl template RNA. All these reagents mix and make a total of 20.04 μl qRT-PCR mixture. It was subjected to a 1-step assay with an applied biosystem model 7,500 instruments under the following conditions: step 1, reverse transcription for 10 min at 55 °C; step 2, for two minutes at 95 °C to activate Taq polymerase; and step 3, denaturation for 10 s at 95 °C; and step 4, annealing and extension at 45 cycles for 60 s at 60 °C. Each positive sample was tested in duplicate with a positive and negative control for cross verification.

## Epidemiological data acquisitions

A structured questionnaire was developed in English. However, it was translated into the local language of that region to increase the accuracy of the respondent rates and decrease the margin of errors to collect the data related to the LBMs and risk factors, that could be associated with the transmission of AIV subtypes (H5N1, H9N2) in humans and poultry.
These questions were about (i) type of poultry; (ii) LBM trading category; (iii) chicken population in LBM; (iv) days operational per week; (v) keeping birds outside cages; (vi) chicken breeds other than broiler in the cages; (vii) rodents in the stall; (viii) mixing the birds from different sources; (ix) sharing of the equipment; (x) mix the poultry arriving on different days; (xi) inspection team from authorities come to visit LBM; (xii) use of detergent during cleaning the wooden tables; (xiii) disinfect the vehicle during deliveries; (xiv) wild birds present around stall; (xv) presence of sick birds in the stall; (xiv) dispose of the dead birds; (xvi) stray dogs accessing the stall; (xvii) presence of ducks in the stall; (xviii) presence of guinea fowl; (xix) presence of turkeys in the stall; (xx) presence of pheasants; (xxi) presence of quails in the stall; (xxii) presence of chukars/partridges in the stall; (xxiii) presence of peafowl in the stall; (xxiv) presence of pet birds in the stall; (xxv) stray cats in the stall; (xxvi) market vehicle picks up birds from the farm. The data collection tools were adapted from previously published questionnaires for a study on AIV risk variables conducted by *Chaudhry et al. (2018)*. The information was gathered *via* hard proformas and personal interviews conducted by the first author in the native language, *i.e.*, Bahasa Indonesia, with the help of veterinary staff from the Animal Husbandry Department and Livestock Services in East Java Province, Indonesia. The study was explained to the poultry employees before they filled out the questionnaire, and verbal consent to participate in the study was acquired. All of the information was meticulously entered into Microsoft Excel sheets and statistically examined.

## Statistical analysis

Statistical analyses were conducted using IBM SPSS version 25. Samples positive for HA and HI were selected for the final detection of qRT-PCR. City and poultry (backyard, broiler, and layer) level prevalence was calculated by dividing the number of positive cases by the total number of samples screened. A multivariable logistic regression analysis was performed to determine the effect of various potential variables on the prevalence of avian influenza H5N1 and H9N2 viruses. We did this by excluding any variables that were used in the same way across all marketplaces, *e.g.*, "working days were excluded as all the markets were working on 7 days." Dummy variables were used to incorporate categorical variables with more than two levels. A dummy variable adjustment method was used to deal with missing data on predictor variables in regression analysis (*Cohen, West & Aiken, 2014*; *Vanden Oord, 2006*). In a multivariable model, all potential variables were added in an additive mode.

To test the significant effect of associated risk factors, two distinct multivariable models were created, one for the H9N2 and the other for the H5N1. The initial multivariable model for H5N1 and H9N2 included 22 variables. In the final models, the response variable (H5N1 and H9N2 outcomes) was fitted as positive or negative. Using the drop1 function, nonsignificant variables ($p > 0.05$) were eliminated one by one, beginning with the highest $p$-value and continuing until the remaining variables had a $p < 0.05$. The "logit" link function was used to determine (1) the coefficient, (2) the coefficient's standard error, and (3) the $p$-value. The exp function was used to determine the odds ratios (OR) and 95

percent confidence intervals (CI) for the final model, as described in a previously published study by *Khan et al. (2021)*.

## RESULTS

### LBMs demographics

A total of eight LBMs were selected in the study from the four cities of East Java, all of which were located in urban areas. When we asked about trading patterns, 93.7% (562) of the respondents said LBMs were a mix of retail and wholesale, while 5% (30) said it was only retail. Only 1.3% (8) of respondents answered that these are wholesale. When questioned about working days, all the respondents stated that LBMs were operated daily, with the same vendors operating the same stalls. When asked about the chicken population in the LBMs, 98% (588) of respondents indicated a high population, 1.5% (nine) a low population, and 0.5% (three) a medium population.

Table 1 enumerates all of the samples taken from each city, the number of visits, name, and date of the month, mixed cases as well as the number that tested positive for HA and HI. Antisera against H5N1 and H9N2 were acquired from Pusat Veteriner Farma in Surabaya, Indonesia, and used for these testing. It was determined that the highest number of positive cases was reported in the rainy season (April–October) as compared to the dry season (November–March) throughout the research. These findings shed light on how the prevalence of avian influenza fluctuates with the seasons (Table 1). Results were considered positive for both subtypes if Ct <30 (Figs. 2 and 3).

A total of 600 tracheal and cloacal swab samples from the poultry (44.5% (267) backyard, 32.83% (197) broiler, and 25.66% (154) layer) of LBMs were collected and screened by HA and HI. The positive HI was further analyzed by using qRT-PCR. 23/600 (3.8%; 95%CI [0.0229–0.0537]) samples were positive for avian influenza A virus subtypes H5N1 and H9N2. The H5 subtype (14 (2.3%)) was detected more frequently than the H9 subtype (9 (1.5%)). 8/23 (34.78%) of samples were positive for both subtypes. The backyard had the highest prevalence of 16/267 (5.99%), followed by broiler 4/179 (2.23%), and layer 3/154 (1.68%). LBMs of Sidoarjo showed a maximum prevalence of 7/150 (4.67%), followed by Surabaya 6/150 (4%), Pasuruan 6/150 (4%), and Malang 4/150 (2.67%). For both subtypes (H5N1 and H9N2), among positive samples, the prevalence was higher for oropharyngeal samples than cloacal samples in all surveyed poultry types (Table 2, Fig. 4).

### Risk factors for H9N2 prevalence in live bird markets

We identified and assessed risk factors for AIV (H9N2) prevalence in LBMs. Twenty-four questionnaires were screened in the multivariable analyses; among them, five factors were significantly associated ($p$-value <0.05) with the outcomes (positive and negative) of H9N2 among the poultry of live bird markets. Logistic regression analysis predicts that the risk of H9N2 was higher among the poultry of live bird markets in the presence of the following potential risk factors: the presence of ducks ($p = 0.003$), turkeys ($p = 0.025$ and pheasants ($p = 0.024$) in the stall, dry ($p = 0.006$) and the rainy season ($p = <0.001$), and birds from the household sources ($p = 0.04$) showed a significant association with the prevalence of avian influenza A virus subtype H9N2 in LBMs. Nineteen factors were found to be

Table 1 Summary of allantoic fluid using HA and HI for isolation of H5 and H9.

| Name of cities with visits | Name of visiting markets | Season and date of samples | Approximate number of stalls/markets | Total samples | HA positive | HI positive H5 | HI positive H9 | Mixed cases (H5, H9) |
|---|---|---|---|---|---|---|---|---|
| Surabaya 1st visit | Pasar Wonokoromo | Dry season 20/05/21 | 20 | 123 | – | – | – | – |
| Surabaya 2nd visit | Tembok market | Rainy season 20/02/2022 | 15 | 27 | 6 | 3 | 3 | 3 |
| Sidoarjo 1st visit | Kacamatan Taman and Krian | Dry season 20/07/21 | 16+10 | 122 | 2 | 1 | 1 | 1 |
| Sidoarjo 2nd visit | Larangan market | Rainy season 24/01/2022 | 12 | 28 | 7 | 2 | 3 | 2 |
| Pasuruan 1st visit | Pandan traditional market | Dry season 04/05/21 | 16 | 123 | 1 | 1 | – | – |
| Pasuruan 2nd visit | Pandan traditional market | Rainy season 14/12/21 | 16 | 27 | 5 | 3 | 2 | 2 |
| Malang 1st visit | Splinded +Kebalen + Blimbing bird market | Dry season 05/06/21 | 13+7+5 | 125 | 2 | 2 | – | – |
| Malang 2nd visit | Kebalen bird market | Rainy season 10/01/2022 | 13 | 25 | 4 | 2 | – | – |

Notes.
HA, Haemagglutination; HI, Haemagglutination inhibition.

protective (*i.e.*, having a *p*-value >0.05) of exposure in the LBMs with the prevalence of avian influenza A virus subtype H9N2 infection (Table 3).

## Risk factors for H5N1 prevalence in live bird markets

The study Table 4 describes the potential risk factors significantly associated with the exposure of avian influenza A virus subtype H5N1 among poultry of live bird markets. A total of 24 variables were screened in the multivariable analyses among them seven factors showed significant association (*p*-value <0.05), while the remaining 17 variables act as a protective factor *i.e.*, *p*-value >0.05. Rodents seen in the stall ($p = 0.03$), stray dogs accessing the stall ($p = 0.04$), presence of ducks ($p = 0.061$), turkeys ($p = 0.03$), chukars/partridges ($p = 0.024$), and presence of peafowl in the stall ($p = 0.003$), rainy season ($p = 0.001$) and birds from the household sources ($p = 0.002$) showed significant relationship ($p < 0.05$) with avian influenza A virus subtype H5N1 prevalence in LBMs (Table 4).

The correlation of associated risk factors and presence or absence of avian influenza infection depicted in Fig. 5.

## DISCUSSION

AI is a zoonotic disease that affects domestic poultry, wild birds, and mammals, including humans. The disease is still endemic in several countries: (Korea, North America, Morocco, (*Suarez, Lee & Swayne, 2006*; *Lee & Song , 2013*; *Essalah-Bennani et al., 2021*), and Egypt (*Hassan, Harder & Hafez, 2021*) including Indonesia (*El Mellouli et al., 2022*; *Rehman et al., 2022a*; *Rehman et al., 2022b*; *Yuniwarti et al., 2012*). LBMs are a significant determinant of avian influenza viruses' transmission because they mix together a diverse variety of

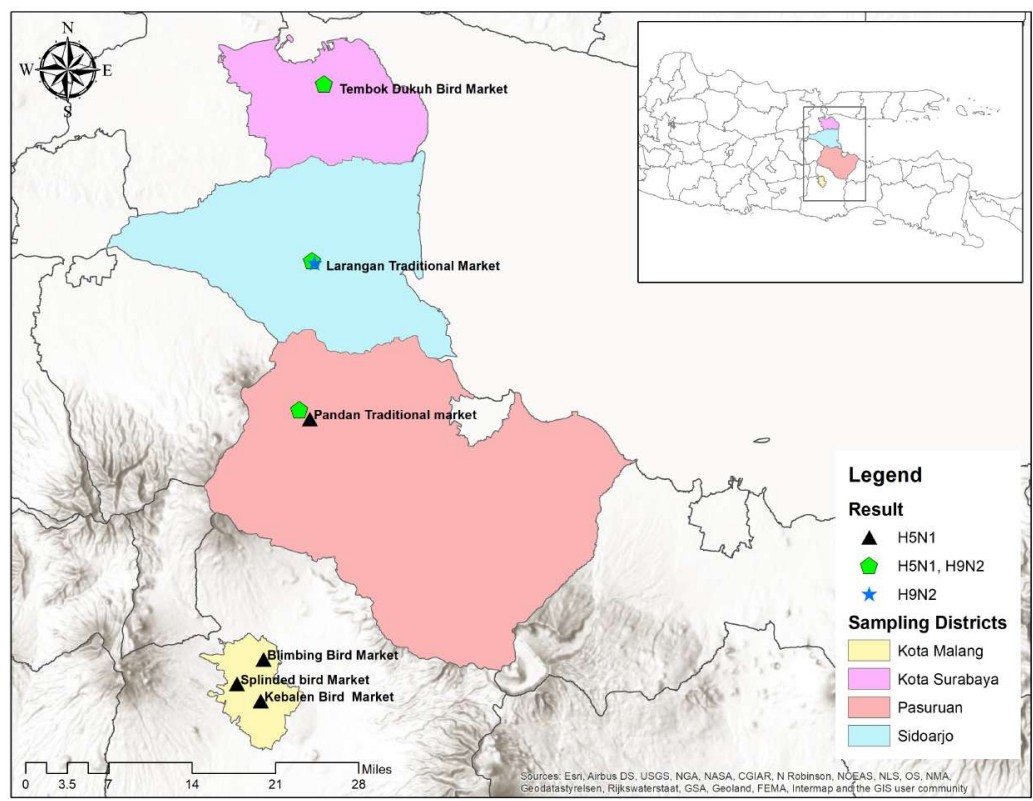

**Figure 2** Estimates of H5N1 and H9N2 infection prevalence in selected LBMs of four cities of East Java.

bird species in a high-density environment conducive to virus genome re-assortment and cross-species dissemination (*Fournié et al., 2012*). Although these LBMs have been identified as prospective "hotspots" for IAVs, their impact on human illnesses remains unknown (*Nguyen et al., 2014*).

We detected avian influenza A virus subtypes H5N1 and H9N2 among the poultry of LBMs of four cities of East Java, Indonesia. The prevalence of the H5N1 virus was higher than the prevalence of the H9N2 virus. The prevalence reported in our study was varied among chickens' maximum prevalence was reported in the backyard as compared to broiler and layer. The higher prevalence of AIVs in the backyard than in the broiler and layer might result from differences in the structure of their respective value chains. The prevalence reported in our study was comparable to the prevalence reported in China (2.5%) (*Wang et al., 2015*), however, it is lower than the prevalence recorded in other countries such as 16.5% in Bangladesh (*Negovetich et al., 2011*), 12.4% in Egypt (*Abdelwhab et al., 2010*), 32.2% in Vietnam (*Nguyen et al., 2014*) and 62.5% in Pakistan (*Rehman et al., 2021*).

The prevalence of AIV at the bird level was lower in our study than in other endemic countries (*Negovetich et al., 2011*; *Nguyen et al., 2014*; *Thuy et al., 2016*). Differences in prevalence estimates between countries could be due to a range of risk factors, such as how birds are handled, maintained, and slaughtered in LBM, the frequency of virus circulation,
# qRT-PCR Curve

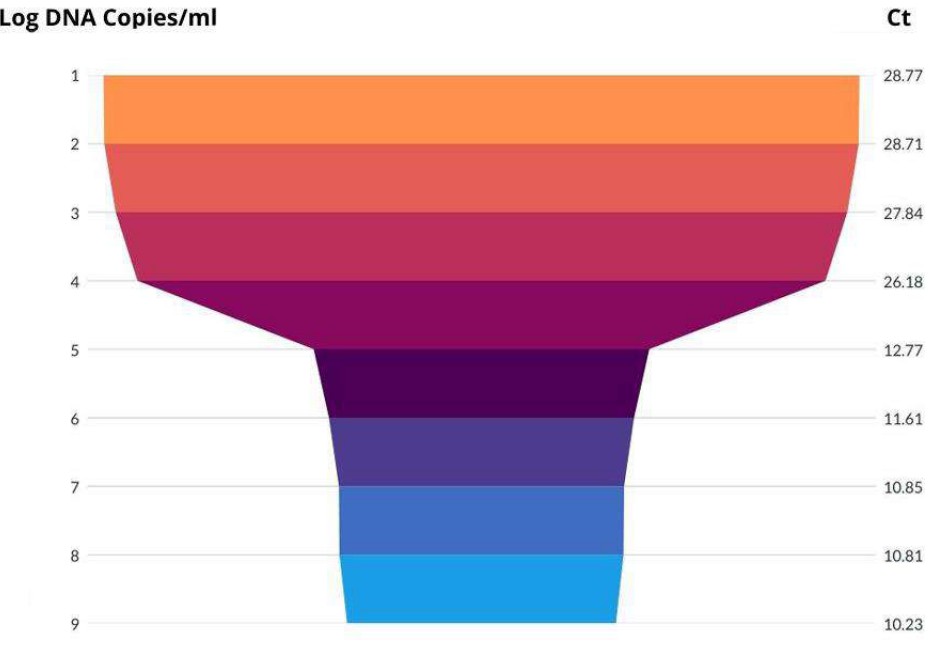

**Log DNA Copies/ml**                                                              **Ct**

| | |
|---|---|
| 1 | 28.77 |
| 2 | 28.71 |
| 3 | 27.84 |
| 4 | 26.18 |
| 5 | 12.77 |
| 6 | 11.61 |
| 7 | 10.85 |
| 8 | 10.81 |
| 9 | 10.23 |

**Figure 3   Graphical representation of H9N2 Ct values.**

survey technique differences, diagnostic methodologies, ecological heterogeneity in study locations, the type of chicken population under observation (backyard poultry, broilers, layers, or wild birds), and the type of study location (Zoo or LBMs).

In the current study, Larangan traditional market Sidoarjo had a higher city-level prevalence than the other studied cities. The greater prevalence estimate could be attributable to the virus's endemicity among commercial and backyard chicken populations in the area (Blitar), from where all poultry for sale in Sidoarjo LBMs comes. Previous research has also revealed that the AI virus subtype H5 was discovered in Sidoarjo' Larangan traditional market (*Frederika et al., 2013*; *Novitasari & Anwar, 2020*).

The higher proportion of backyards marketed in assessed LBMs could indicate that backyards are more likely to be sourced from large flocks than broiler and layer chickens, which are subsequently mixed in densely populated trucks during transport to LBMs, boosting AIV transmission. These patterns of occurrence could be due to differences in levels of genetic vulnerability to AIV infection (*Blohm et al., 2016*; *Ruiz-Hernandez et al., 2016*).

More research is needed to separate the effects of trade-related and genetic factors on AIV transmission in these chicken types (*Fournié et al., 2017*). The cohabitation of avian influenza A virus subtypes H5N1 and H9N2 raises concerns about the emergence of novel recombinant strains (*Gao et al., 2013*; *Lin et al., 2000*). The presence of both subtypes in

Rehman et al. (2022), *PeerJ*, DOI 10.7717/peerj.14095

**Table 2  Prevalence of avian influenza A viruses (H5 and H9) in poultry using qRT-PCR approach in four cities of East Java Indonesia.**

| Name of cities | Backyard | | | | Broiler | | | | Layer | | | |
|---|---|---|---|---|---|---|---|---|---|---|---|---|
| | Collected samples | H9 positive (%) | H5 positive (%) | Total positive (%) | Collected samples | H9 positive (%) | H5 positive (%) | Total positive (%) | Collected samples | H9 positive (%) | H5 positive (%) | Total positive (%) |
| Surabaya | 56 | 3 (5.36%) | 3 (5.36%) | 6 (10.71%) | 59 | – | – | – | 35 | – | – | |
| Sidoarjo | 89 | 3 (3.37%) | 3 (3.37%) | 6 (6.74%) | 48 | – | – | – | 13 | 1 (7.69%) | – | 1 (7.69%) |
| Pasuruan | 50 | 1 (2%) | 1 (2%) | 2 (4%) | 33 | 1 (3.03%) | 2 (6.06%) | 3 (9.09%) | 67 | – | 1 (1.49%) | 1 (1.49%) |
| Malang | 72 | – | 2 (2.77%) | 2 (2.77%) | 39 | – | 1 (2.56%) | 1 (2.56%) | 39 | – | 1 (2.56%) | 1 (2.56%) |
| Total | 267 | 7 (2.62%) | 9 (3.37%) | 16 (5.99%) | 179 | 1 (0.5%) | 3 (1.68%) | 4 (2.23%) | 154 | 1 (0.64%) | 2 (1.30%) | 3 (1.94%) |

# qRT-PCR Curve

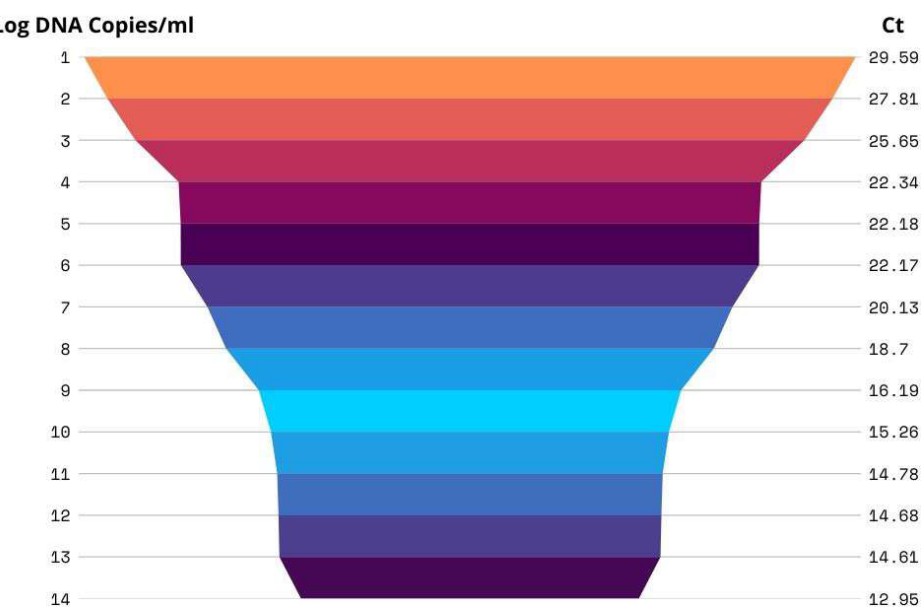

**Log DNA Copies/ml**                                                **Ct**

| | |
|---|---|
| 1 | 29.59 |
| 2 | 27.81 |
| 3 | 25.65 |
| 4 | 22.34 |
| 5 | 22.18 |
| 6 | 22.17 |
| 7 | 20.13 |
| 8 | 18.7 |
| 9 | 16.19 |
| 10 | 15.26 |
| 11 | 14.78 |
| 12 | 14.68 |
| 13 | 14.61 |
| 14 | 12.95 |

**Figure 4** **Graphically representation of H5N1 Ct values.**

**Table 3** **Summary of the potential risk factors associated with prevalence of avian influenza subtype H9N2 in live bird markets of East Java, Indonesia.**

| Variables | Response categories | Response n (%) | OR | 95% CI | p-value |
|---|---|---|---|---|---|
| Presence of ducks in the stall. | No | 249 (41.5%) | reference | | 0.013 |
| | Not Sure | 160 (26.7%) | 38.02 | 3.404–430.10 | 0.003 |
| | Yes | 191 (31.8%) | <0.0001 | – | 0.995 |
| Type of poultry in the flock | Layer | 154 (25.7%) | reference | | 0.085 |
| | Broiler | 179 (29.8%) | 6.957 | 0.744–65.089 | 0.089 |
| | Backyard | 267 (44.5%) | 1.062 | 0.058–19.295 | 0.995 |
| Presence of turkey in the stall. | No | 252 (42%) | reference | | 0.032 |
| | Not Sure | 203 (33.8%) | 0.028 | 0.001–0.633 | 0.025 |
| | Yes | 145 (24.2%) | 0.032 | 0.002–0.540 | 0.017 |
| Presence of pheasants in the stall. | No | 243 (40.5%) | reference | | 0.073 |
| | Not Sure | 231 (38.5%) | 14.601 | 0.640–333 | 0.093 |
| | Yes | 145 (24.2%) | 18.422 | 1.471–231 | 0.024 |
| Effect of season | Dry season | 493 (82.2%) | reference | – | 0.006 |
| | Rainy season | 107 (17.8%) | 0.05 | 0.025–0.075 | <0.001 |
| Sources of birds | Household | 267 (44.5%) | reference | | 0.042 |
| | Farms | 333 (55.5%) | 0.224 | 0.046–1.089 | 0.064 |

**Notes.**

   CI, Confidence interval; OR, Odd ratio.

**Table 4 Summary of the significant potential risk factors associated with prevalence of avian influenza subtype H5N1 in live bird markets of East Java, Indonesia.**

| Variables | Response categories | Response n (%) | OR | 95% CI | *p*-value |
|---|---|---|---|---|---|
| Do you see rodents in the stall? | No | 181 (30.2%) | reference | | 0.05 |
| | Not sure | 164 (27.3%) | 0.005 | <0.0001–0.706 | 0.036 |
| | Yes | 255 (42.5%) | 115.152 | 0.337–39318 | 0.111 |
| Do you mix birds from different sources? | No | 187 (31.2%) | reference | | 0.03 |
| | Not sure | 154 (25.7%) | 16.10 | 0.426–678 | 0.132 |
| | Yes | 259 (43.2%) | 0.021 | <0.0001–13.94 | 0.145 |
| Stray dogs accessing the stall | No | 237 (39.5%) | reference | | 0.04 |
| | Not sure | 209 (34.8%) | <0.001 | <0.0001–0.173 | 0.014 |
| | Yes | 154 (25.7% | 0.123 | 0.002–4.551 | 0.24 |
| Presence of ducks in the stall | No | 249 (41.5%) | reference | | 0.174 |
| | Not sure | 203 (33.8%) | 17.037 | 0.874–332 | 0.061 |
| | Yes | 145 (24.2%) | <0.0001 | <0.0001–0.0001 | 0.10 |
| Presence of turkey in the stall | No | 252 (42%) | reference | | 0.096 |
| | Not sure | 203 (33.8%) | 0.141 | 0.004–4.58 | 0.270 |
| | Yes | 145 (24.2%) | 0.007 | <0.001–0.629 | 0.031 |
| Presence of chukars/-partridges in the stall. | No | 229 (38.2%) | reference | | 0.073 |
| | Not sure | 224 (37.3%) | 2500 | 2.802–2232 | 0.024 |
| | Yes | 147 (24.5%) | 0.818 | 0.013–52.47 | 0.925 |
| Presence of pea fowl, in the stall. | No | 245 (40.8%) | reference | | 0.010 |
| | Not sure | 190 (31.7%) | <0.001 | <0.001–0.032 | 0.003 |
| | Yes | 165 (27.5%) | 1.175 | 0.031–44.773 | 0.931 |
| Presence of pet birds, in the stall. | No | 204 (34%) | reference | | 0.118 |
| | Not sure | 143 (23.8%) | 20.15 | 0.759–535 | 0.073 |
| | Yes | 253 (42.2%) | 1.18 | 0.036–38.87 | 0.925 |
| Effect of season | Dry season | 493 (82.2%) | reference | | 0.062% |
| | Rainy season | 107 (17.8%) | 0.130 | 0.20–0.083 | 0.001 |
| Sources of birds | Household | 267 (44.5%) | reference | | 0.002 |
| | Farms | 333 (55.5%) | 0.08 | 0.145–1.320 | 0.142 |

**Notes.**
    CI, Confidence interval; OR, Odd ratio.

some poultry samples shows that they coexisted or shared a host during the research period (*Hulse-Post et al., 2005*).

## Risk factors associated with the prevalence of avian influenza

The analysis of risk factors presented in this study suggests that the presence of ducks, turkeys, chukars/partridges, peafowl, and pheasants in the stall is strongly associated with an increased risk of AIV subtypes H5N1 and H9N2 in live bird markets of the study province. In previous studies, ducks in LBMs were found to be a risk factor for AIVs, whereas backyard duck concentrations were not linked. Although the existence of ducks as an associated risk in rice-rotation settings cannot be ruled out, ducks are expected to have less impact on HPAI H5N1 circulation in Indonesia than in other Southeast Asian countries (*Gilbert et al., 2006*; *Indriani et al., 2010*; *Loth et al., 2011*; *Tiensin et al., 2007*).

| Name of variable | H5N1 | H9N2 |
|---|---|---|
| Type of poultry in the flock | -0.03196 | -0.07231 |
| Live bird markets trading category | -0.03277 | 0.03030 |
| Chicken population in LBMs | 0.02142 | 0.01710 |
| Keeping birds outside cages | -0.01898 | -0.0301 |
| Chicken breeds other than 2 in the cages | -0.00393 | -0.06115 |
| Sight of rodents in the stall | -0.00357 | -0.03072 |
| Mix birds from different sources | -0.05586 | -0.06295 |
| Shared slaughtering, feeding, watering, and weighing equipment | -0.04314 | -0.04264 |
| Arrival of mix poultry on different days in same cages | -0.01492 | -0.01427 |
| Visits of LBMs by the govt Inspection authorities | -0.02157 | -0.03216 |
| Use of detergent for cleaning the wooden tables | -0.03156 | -0.03948 |
| Mixing of different species in same cage | -0.02933 | -0.0376 |
| Disinfection of the vehicle during deliveries from different areas | 0.03849 | -0.04578 |
| Presence of wild birds around stalls | 0.02791 | -0.03127 |
| Presence of sick birds in stalls | 0.0291 | -0.02325 |
| Dispose the dead birds | 0.01338 | -0.00080 |
| Access of stray dogs to the stalls | 0.08417 | 0.06473 |
| Presence of ducks in the stalls | -0.0175 | -0.03013 |
| Presence of Guinea fowl in the stalls | -0.02603 | -0.01445 |
| Presence of Turkey in the stalls | 0.00700 | 0.05865 |
| Presence of Pheasants in the stalls | -0.01061 | -0.01362 |
| Presence of Quails in the stalls | -0.00706 | -0.01045 |
| Presence of Chukars/partridges in the stalls | 0.00122 | -0.00404 |
| Presence of Pea fowl in the stalls | 0.0694 | 0.01344 |
| Presence of Pet birds in the stalls | 0.00182 | -0.06730 |
| Presence of Stray cats in the stalls | -0.00423 | -0.081 |

**Labels (each color show correlation)**

| Correlation scale | Relationship |
|---|---|
| 1 | A perfect +ev |
| 0.7 | Strong +ev |
| 0.5 | Moderate +ev |
| 0.3 | A weak +ev |
| -1 | A perfect -ev |
| -0.7 | Strong -ev |
| -0.5 | A moderate -ev |
| -0.3 | A weak -ev |
| 0 | No correlation |

**Figure 5** Heatmap depicting the correlation between the risk factors and presence or absence of infection of H5N1 and H9N2.

A previous study in Pakistan found that due to poor biosecurity, the risk of AIV is increased in these multi-species markets. Mixing birds from various sources may potentially contribute to the spread of AIV in these stalls (*Abbas et al., 2011*; *Chaudhry et al., 2017*). An earlier study in Indonesia concluded that the source of stall birds posed no danger of AIV infection. However, environmental samples were used instead of chickens being swabbed directly in this study (*Indriani et al., 2010*; *Loth et al., 2011*). However, in the current study, this factor was found to have a strong relationship with the prevalence of AIVs, demonstrating the uniqueness of the study risk variables. In the current scenario, H5N1 infection was shown to be considerably more correlated with stray dogs accessing the stalls ($p = 0.014$) than stray cats visiting the stalls. As a result, we recommend always being vigilant about the entrance of stray dogs near the stalls to avoid H5N1 infection.

The presence of chukars/partridges in the stall also showed a significant association with H5N1 infection in LBMs ($p = 0.04$). Previously, LBMs in Pakistan revealed no link between this factor and AIV infection (*Chaudhry et al., 2018*). In a multivariable analysis, the presence of rodents in the stall ($p = 0.036$) indicated a significant positive connection with AI subtype H5N1, but no association with AIV H9N2. A pest management program in LBMs could help to prevent AIV transmission via this pathway. The influence of closing days could not be assessed because all LBMs in our study were open seven days a week. Other research has discovered that this lowers the chance of AI infection in LBMs (*Ma et al., 2014*). According to our study finding birds sample during the rainy season (November–March) had a significant association with the prevalence of AIVs as compared to dry

weather (April–October). A comparable seasonal effect was observed in the tropics during an investigation of native chickens in Bangladesh (*Nooruddin et al., 2006*). In the current investigation, no association was discovered between several other parameters (LBMs trading category, the chicken population in LBMs, days operational per week, keeping birds outside cages, sharing of the equipment, mixing the poultry arriving on different days, an inspection team from authorities coming to visit LBMs, use of detergent during cleaning the wooden tables, disinfecting the vehicle during deliveries, wild birds present around the stall, presence of sick birds in the stall, disposal of the dead birds, presence of quails in the stall, and stray cats in the stall) and H5N1 and H9N2 infection. These variables, however, have been found as significant determinants in other investigations (*Cappelle et al., 2014*; *Cardona, Yee & Carpenter, 2009*; *Wang et al., 2017*; *Wu et al., 2014*; *Zhou et al., 2016*).

## Study limitations

Our study had several limitations. As it was a cross-sectional study, in which samples were gathered over a short period to minimize variability caused by seasonal fluctuations in AIV prevalence. As a result, our estimations solely covered AIV prevalence in that exclusive period and did not account for seasonal variations. Seasonal fluctuations like humidity, temperature, and precipitation are essential predictors to be considered in long-term epidemiological studies. Because our study was based on cross-sectional data, causation could not be determined and the risk variables for AIV prevalence could not be completely explored. As a result, more robust analytical and epidemiological studies are needed to further explore the different strains of AIVs in detail. Unlike prior cross-sectional research, our method allowed us to estimate AIV prevalence not only by poultry species but also by chicken type, as well as the type of LBMs in which the sampled birds were sold. The market administrators' and sellers' reports about inspection and cleaning practices in LBMs were not validated during the investigation. These practices may have been exaggerated because participants tried to tell interviewers what they believed they wanted to hear.

## CONCLUSIONS

The aim of this study was to determine the prevalence of AI H5N1 and H9N2 subtypes in LBMs and to identify potential risk variables associated with AI infections in Indonesia to help policymakers and other concerned authorities with effective control and eradication strategies. AI subtypes H5N1 and H9N2 infections in LBMs were confirmed by qRT-PCR, supporting the hypothesis that both subtypes were propagating and perhaps lingering in these endemic spots. The findings of this study also reveal a number of risk factors that are similar to those found in previous studies conducted in different countries, including, Egypt, Vietnam, Malaysia, Afghanistan, China, and India, indicating that these common factors are the primary cause of increasing AI (H5 and H9) infections. Controlling these risk variables could help to minimize the prevalence of AIV in Indonesia, particularly in the research area. Continuous AI monitoring of LBMs in major cities could help researchers and other stakeholders avert perspective epidemics.

## ACKNOWLEDGEMENTS

We are grateful for the assistance of the veterinarians, technicians, the LBMs owners, and workers. The Virology and Biomolecular Laboratory staff, Faculty of Veterinary Medicine Universitas Airlangga, Animal Husbandry Department, and Livestock Services East Java Province, Indonesia were acknowledged for their cooperation in the study.

### Funding

Logistic support for field sampling, consumables required for laboratory work and APCs were fully funded by the Penelitian Hibah Mandat funding from Universitas Airlangga, Indonesia in the fiscal year 2022, with grant number: 220/UN3.15/PT/2022. There was no additional external funding received for this study. The funders had no role in study design, data collection and analysis, decision to publish, or preparation of the manuscript.

### Grant Disclosures

The following grant information was disclosed by the authors:
Universitas Airlangga, Indonesia: 220/UN3.15/PT/2022.

### Competing Interests

The authors declare there are no competing interests.

### Author Contributions

- Saifur Rehman conceived and designed the experiments, performed the experiments, analyzed the data, prepared figures and/or tables, authored or reviewed drafts of the article, and approved the final draft.
- Mustofa Helmi Effendi conceived and designed the experiments, performed the experiments, analyzed the data, prepared figures and/or tables, authored or reviewed drafts of the article, and approved the final draft.
- Aamir Shehzad conceived and designed the experiments, prepared figures and/or tables, and approved the final draft.
- Attaur Rahman analyzed the data, prepared figures and/or tables, authored or reviewed drafts of the article, and approved the final draft.
- Jola Rahmahani conceived and designed the experiments, performed the experiments, prepared figures and/or tables, and approved the final draft.
- Adiana Mutamsari Witaningrum performed the experiments, prepared figures and/or tables, and approved the final draft.
- Muhammad Bilal conceived and designed the experiments, prepared figures and/or tables, and approved the final draft.

### Ethics

The following information was supplied relating to ethical approvals:

The Animal Care and Use Committee at the Faculty of Veterinary Medicine, Universitas Airlangga Surabaya, Indonesia (Approval no. 1.KE.028.03.2021) gave their approval for every step of this study.

## Data Availability

The data is available at Figshare: Rehman, Saifur; Rantam, Fedik Abdul; Helmi effendi, Mustofa; Shehzad, Aamir; Rahman, Attaur; Rahmahani, Jola; et al. (2022): Prevalence and associated risk factors of avian influenza A virus subtypes H5N1 and H9N2 in LBMs of East Java Province, Indonesia: A cross-sectional study. figshare. Dataset. https://doi.org/10.6084/m9.figshare.20000117.v1.

## Supplemental Information

Supplemental information for this article can be found online at http://dx.doi.org/10.7717/peerj.14095#supplemental-information.

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
