# Peer review of "Prevalence and associated risk factors of avian influenza A virus subtypes H5N1 and H9N2 in LBMs of East Java province, Indonesia: a cross-sectional study"

_PeerJ, doi:10.7717/peerj.14095_

## Round 0.1 · original submission · Major Revisions

Thank you for submitting your manuscript to PeerJ. I have received comments from reviewers on your manuscript. Your paper should become acceptable for publication pending suitable MAJOR revisions and modification of the article in light of the appended reviewer comments.

Reviewer 1 ·

Basic reporting

The manuscript entitled “Prevalence and associated risk factors of avian influenza A virus subtypes H5N1 and H9N2 in LBMs of East Java Province, Indonesia: A cross-sectional study” describe the prevalence and the risk factors associated with H5N1 and H9N1 prevalence in wet and LBM Java Province, Indonesia from March 2021 to April 2022.
The professional English used is clear, unambiguous and technically correct . with sufficient literature background and references. However, one of the main drawbacks of the study is the inefficient data visualization. suitable figures are required to show the prevalence and the risk factors correlation. also, it is not clear for me what is the significance of the PCR figures in the context of the manuscript. Finally in this regard, I can't find any legends for tables and figures!!

Experimental design

the research question is well defined, relevant and meaningful. However, it missed very important steps that limited the importance of the out come as the absence of gene sequencing of the positive samples and environmental samples (around the birds). further, absence of the correlation of positive cases with the season of samples collection. also, absence of information about the source of the birds that can reflect the immune status.
Regarding methods used, the virus isolation and detection section was not fully described. it was not clear why they preferred to isolate viruses before applying PCR, this approach reduce the sensitivity of detection system. also, it was not clear why authors used HI assay and why only antisera raised against specific H5N1 strain.
The analysis section can be improved to have more easily visualized data out of the collected questionnaires. also, more idea for analysis should be used for example classifying questions into bundles and each one should be correlated to the presence or absence of infection in separate figure (using heat map will be useful).

Validity of the findings

Despite the topic of great importance, the absence of gene sequencing of the positive samples and environmental samples (around the birds) limiting the importance of the outcome. Further, the analysis of risk factors should include the season factor during study time (March 2021 to April 2022) and the source of birds (household or farms), unless broiler and layer that you sampled only come from commercial farm that need to be clarified. Variables should be classified into bundles and each one should be correlated to the presence or absence of infection in separate figure (using heat map will be useful).

Additional comments

1. It is not clear whether there is demarcation between wet and LBM, if yes can you specify and separate the results of each one. If not make it clear in the study population and sampling section.
2. Some information about the vaccination policy in Indonesian commercial and backyard farms, and correlation to disease prevalence should be added to the introduction section.
3. LINE 85-89: not clear
4. The number ‘approximate’ of stalls should be included in each market and date of sampling.
5. Line 129: clarify the reason for injecting antibiotics separately and add reference for this approach, since you used the antibiotic in the viral transport media.
6. In files that I received there was not legend for figures and tables.
7. What is the significance of showing the PCR Figures?
8. What is the reason of using HI technique?
9. Line 133: why only H5N1 (clade. 2.3.2) antisera used? do you have sequence data or used specific PCR?
10. Include the mixed infection cases in table 1.

Reviewer 2 ·

Basic reporting

Comments to the author(s)
Rehman et al investigated the prevalence and associated risk factors of avian influenza A
virus subtypes H5N1 and H9N2 in LBMs of East Java Province, Indonesia. Such epidemiological studies are needed to follow the epidemiology of avian
influenza. I recommend the publication of this manuscript after major revision.
Major comments
1-The manuscript needs a language editing by native Speaker
2- Please use the abbreviation of avian influenza virus (AIV) throughout the manuscript
3- Line 23: avian influenza subtypes H5N1 and H9N2 (use this nomaclature throughout the manuscript)
Line 31: should be: The samples were inoculated into specific pathogenic free emberyonated eggs at 9-day-old via allantoic fluid
Line 41: Please expand the full name of LBMs?
Line 84: please expand the full name of HPAI
Line 124: Virus isolation
Line 243: please use the abbreviation of AI
Line 144: The disease is still endemic in several countries (https://doi.org/10.51585/gjvr.2021.2.0012 and https://doi.org/10.51585/gtop.2021.0004)
including Indonesia...........................
Line 249: please check the citation style?
Line 322:please check the citation style?

Experimental design

no comment

Validity of the findings

The findings are validated

Additional comments

no

Annotated reviews are not available for download in order to protect the identity of reviewers who chose to remain anonymous.

·

Basic reporting

this manuscript is written in clear language. introduction and discission are supplemented with sufficient references and background information. the study design is appropriate with clear methodology and results. conclusion is supported by the results

Experimental design

the Experimental design is clear and appropriate, and fills the gap of knowledge regarding the risk factors related with avian influenza speeding through live bird market, and thus its impact of the possibility of coinfection and reassortment between different subtypes, along with the possibility of acquiring mutations and mammalian adaptation markers.
questionnaires acquired sufficient information and analyzed properly.
however, the authors tested all collected samples for only H5N1 clade 2.3.2 and H9N2 viruses. they collected the samples and subjected them to propagate in the allantois fluid of SPF eggs, then tested the harvest with HA assay then HI against antibodies specific to only H9N2 and H5N1 (clade. 2.3.2) before testing by RT-PCR. this means that any other subtype that may present or other clades of H5 will be missed.
also, the RT-PCR mentioned is specific to HA subtyping, and not NA, and then claimed the NA subtyping with no mentioned PCR or confirmatory test . are authors sure that no other H5 than H5N1 is circulating? no H5N6, H5N2 or H5N8?
i would also preferer if authors mentioned the numbers of visits and in which months, were all markets in different areas visited in same season?? that is directly affecting the percent of positivity among the samples.

Validity of the findings

findings are valid and important to know, all the claims are supported with strong analysis in my opinion.
more findings could be acquired from current great amount of data collected. for example, the positivity through months/ sites.

Additional comments

generally, this manuscript is very informative and was written in clear language. all conclusions are supported by the great amount of collected data and appropriate statistical analysis.

---

## Round 0.2 · accepted · Accept

Reviewer #3 made a minor comment, which should be addressed in the final version of the manuscript.

Reviewer 2 ·

Basic reporting

The manuscript is improved , thanks for authors

Experimental design

No further comment

Validity of the findings

No further comment

Additional comments

No further comment

·

Basic reporting

the authors made great effort editing the manuscript. they replied to all comments by reviewers , and clarified the missing points.

Experimental design

line 225-226 please add reference to the primers and probes if available

Validity of the findings

was there any difference between detection by HI and PCR in positivity of samples